



# Seasonal evaluation of tropospheric $CO_2$ over the Asia-Pacific region observed by the CONTRAIL commercial airliner measurements

Taku Umezawa[1], Hidekazu Matsueda[2], Yousuke Sawa[2], Yousuke Niwa[2], Toshinobu Machida[1], and Lingxi Zhou[3]

[1]National Institute for Environmental Studies, Tsukuba, Japan
[2]Meteorological Research Institute, Tsukuba, Japan
[3]Chinese Academy of Meteorological Sciences, Beijing, China

*Correspondence to*: Taku Umezawa (umezawa.taku@nies.go.jp)

**Abstract.** We present climatological carbon dioxide ($CO_2$) distributions over the Asia-Pacific region obtained from the CONTRAIL (Comprehensive Observation Network for Trace gases by Airliner) measurements. The high-frequency in-flight $CO_2$ measurements over 10 years reveal a clear seasonal variation of $CO_2$ in the upper troposphere (UT), with a maximum occurring in April–May and a minimum in August–September. The $CO_2$ mole fraction in the UT north of 40° N is low and highly variable in June–August due to the arrival of air parcels with seasonally low $CO_2$ caused by the summertime biospheric uptake in boreal Eurasia. For August–September in particular, the UT $CO_2$ is noticeably low within the Asian summer monsoon anticyclone associated with the convective transport of strong biospheric $CO_2$ uptake signal over South Asia. During September as the anticyclone decays, a spreading of this low $CO_2$ area in the UT is observed in the vertical profiles of $CO_2$ over the Pacific Rim of the continental East Asia. Simulation results identify the influence of anthropogenic and biospheric $CO_2$ fluxes in the seasonal evolution of the spatial $CO_2$ distribution over the Asia-Pacific region. It is found, for example, that a substantial contribution to the UT $CO_2$ over the northwestern Pacific comes from the continental East Asian emissions in the spring, but switches to South Asian and/or Southeast Asian air masses affected dominantly by the biospheric $CO_2$ uptake in the summer monsoon season. The CONTRAIL $CO_2$ data provide useful constraints to model estimates of surface fluxes and to the evaluation of the satellite observations, in particular for the Asia-Pacific region.

## 1 Introduction

Actions for mitigating climate change require accurate knowledge of global budgets of greenhouse gases. It has been estimated that approximately one-half of $CO_2$ emissions had remained in the atmosphere during the period 1959–2010, with the rest taken up by land and ocean sinks (Ballantyne et al., 2012). With a rapidly growing economy in recent decades, Asia has become increasingly important in the global carbon budget. China is now the world's largest $CO_2$ emitter, with India, Japan, and the Republic of Korea all in the world's top 10 emitting nations (Boden et al., 2016). At the same time, Asia has gone through significant land use and land cover changes, impacting the magnitude and the spatial distribution of terrestrial





carbon fluxes (e.g. Calle et al., 2016; Cervarich et al., 2016). However, there are still large uncertainties in the estimates of every component of the Asian carbon budget.

To estimate surface $CO_2$ fluxes, atmospheric transport models have been conventionally constrained by various surface measurement networks (e.g. Gurney et al., 2002; Patra et al., 2008). But due to the sparseness of the surface measurement

sites in Asia, increasing number of modeling studies that have focused on the Asian carbon budget (e.g. Patra et al., 2011; Niwa et al., 2012; Zhang et al., 2014; Jiang et al., 2014, 2016) in recent years started to incorporate $CO_2$ data taken by commercial airliners, such as CARIBIC (Civil Aircraft for the Regular Investigation of the atmosphere Based on an Instrument Container) (Brenninkmeijer et al., 2007) and CONTRAIL (Comprehensive Observation Network for TRace gases by AIrLiner) (Machida et al., 2008). It has been demonstrated that by incorporating the CARIBIC and CONTRAIL

data, model estimates of the Asian $CO_2$ fluxes have been significantly improved (Patra et al., 2011; Niwa et al., 2012; Shirai et al., 2017).

The dominant seasonally-varying atmospheric circulation regime that has an important influence on the variations of atmospheric trace gases throughout the troposphere over Asia is the monsoon circulation (e.g. Lawrence and Lelieveld, 2010). Seasonal variations in trace gases observed at ground stations, as well as in the upper troposphere (UT), have been

found to be influenced by the monsoon circulation (Xiong et al., 2009; Park et al., 2009; Randel et al., 2010; Schuck et al., 2010). In this study, we focus on some of the less-well studied $CO_2$ distributions in association with the Asian monsoon. In this respect, measurements from commercial airliners that fly in the UT are analyzed to provide invaluable insight into the seasonality of the vertical dynamical connection between atmospheric $CO_2$ and the surface flux.

The CONTRAIL project has obtained high-frequency $CO_2$ measurements along flight tracks, as well as vertical profiles

during the ascent and descent over airports, providing a more comprehensive time-dependent three-dimensional spatial distribution of atmospheric $CO_2$. Analyses of seasonal variations and meridional transport of $CO_2$ in the free troposphere (FT; including the UT) and in the lowermost stratosphere using data from CONTRAIL have been presented by Sawa et al. (2008, 2012). Sawa et al. (2012) analyzed the CONTRAIL $CO_2$ data for the period 2005–2010; the number of flights used in the study exceeded 5000, giving nearly 3 million $CO_2$ measurement values. By the end of 2015, we have more than doubled

the amount of measurement values, allowing us not only to update their results but also to explore additional spatiotemporal $CO_2$ variations. The present study addresses climatological $CO_2$ distributions over the Asia-Pacific region and the influence of Asian surface fluxes under varying seasonal atmospheric conditions, as well as to provide a baseline for future optimal use of the CONTRAIL $CO_2$ data. In Section 2, we describe the CONTRAIL $CO_2$ measurements, as well as data analysis procedures, and model simulations to aid in the interpretation of the observations. In Section 3, we evaluate seasonal

distributions of $CO_2$ in both observation and model data. In Section 4, we discuss three interesting features found by our measurements: the summertime low $CO_2$ associated with the Asian summer monsoon, another low $CO_2$ originating in the boreal summer biospheric uptake, and the springtime high $CO_2$ observed in East Asia. Concluding remarks are given in Section 5.



## 2 Method

### 2.1 Experimental

The CONTRAIL project (http://www.cger.go.jp/contrail/) deploys two types of instruments onboard aircraft:
Continuous $CO_2$ Measuring Equipment (CME) and Automatic air Sampling Equipment (ASE). We refer to Machida et al.
(2008) for details, and only a brief description on the CME unit is given here. The CME measures $CO_2$ mole fractions
onboard the aircraft using a non-dispersive infrared gas analyzer (NDIR; LI-840, LI-COR Biogeosciences). As of May 2018,
installation of the CME is certified for eight Boeing 777-200ER and two Boeing 777-300ER aircraft of Japan Airlines (JAL).
Once installed, the CME is operated automatically using the aircraft's flight navigation data until it is unloaded from the
aircraft two months later. The CME samples air from the air conditioning system on the aircraft. The flow rate and the
absolute pressure of the sample air in the NDIR cell are maintained at a constant level to minimize signal drift. The measured
sample values are compared with two working standard gases ($CO_2$ in air) in high-pressure cylinders (2 L) installed inside
the CME and the measurements are traceable to the NIES (National Institute for Environmental Studies)-09 $CO_2$ scale. Mole
fraction of $CO_2$ in dry synthetic air in $\mu mol\ mol^{-1}$ is reported in ppm in this paper. The latest results from the Round Robin
intercomparison experiment show that the NIES-09 $CO_2$ scale differs from the WMO-CO2-X2007 scale by less than 0.1
ppm (http://www.esrl. noaa.gov/gmd/ccgg/wmorr/wmorr_results.php). The standard gases are currently introduced into the
NDIR cell every 14 min during the ascent/descent portion of the flight and every 60 min during the constant altitude portion
of the flight (cruise) typically at 8–12 km. These standard gas intervals were initially 10 min during ascent/descent and 20
min during cruise until December 2005; the 20-min interval was then changed to 40 min until October–November 2011. The
CME data are recorded as 10-s averaged measurements during ascent/descent (~100-m intervals in altitude) and at 1-min
intervals during cruise (~15 km intervals horizontally). To avoid heavy pollution around airports, the CME is not operated
within 2000 ft (609.6 m) of the ground surface (this altitude was initially set to 1200 ft until March–June 2007). The overall
analytical precision of the CME is estimated to be < 0.2 ppm.

For the 10-year period from 2005 to 2015, we collected > 7 million $CO_2$ data points from > 12 thousand flights all over
the world. The CME measurements over the Asia-Pacific region are shown in Fig. 1. Flights from Japan to Southeast Asia
(Bangkok (BKK), Singapore (SIN) and Jakarta (CGK)) provide measurements over the East China Sea, the South China Sea,
the Indochina Peninsula and the maritime continent. These measurement areas are substantially overlapped by flights to the
continental East Asia (Incheon (ICN), Shanghai (SHA) and Hong Kong (HKG)) and to Taipei (TPE). Flights to Delhi (DEL)
provide a unique opportunity for observations over the continental Asia. In addition, extensive measurements from Japan to
the north, to the east and to the south are achieved from flights to Europe, to the North America and Hawaii, and to Australia,
respectively. The major airports where CONTRAIL CME measurements in Asia are made, along with the number of vertical
profile measurements of $CO_2$ over each airport, are listed in Table 1. Vertical profile data with less than 10 $CO_2$ data points



are not used in this study. As indicated in the table, the largest number of $CO_2$ data has been obtained over the Tokyo Narita (NRT) airport with over 7000 vertical profiles, followed by Tokyo Haneda (HND) with over 3600 profiles. Figure 1b shows the number of monthly vertical profiles taken over the airports listed in Table 1. As seen in this figure, the CME measurements have acquired over 30 vertical profiles per month (colored red; i.e. at least one or multiple ascent/descent

flights every day on average) over NRT and HND. Although measurements are less regular compared to these airports, a substantial number of vertical profiles have also been taken over other airports and those data that cover much of the year are presented in this study.

### 2.2 Data Analysis

In this study, we focus on $CO_2$ variations in the troposphere. Observations in the UT are, however, quite often influenced by stratospheric air that has distinct characteristics in the atmospheric composition (e.g. Hoor et al., 2002; Sawa et al., 2004, 2008, 2015). These data are excluded from the dataset based on potential vorticity (PV) values. PV at the location and time of each $CO_2$ measurement taken by CONTRAIL is calculated from the JCDAS (Onogi et al., 2007) and the JRA-55 (Kobayashi et al., 2015) reanalysis datasets (the latter being used since 2014), and any data accompanied by PV values of > 2

PVU (1 PVU = $10^{-6}$ m$^2$ s$^{-1}$ K kg$^{-1}$) are excluded. It has been found that the 2-PVU criteria is relatively robust in separating out the $CO_2$ measurements in the UT that are stratospherically influenced from those that are not (Sawa et al., 2008, 2015). In total, 33% of the CONTRAIL CME $CO_2$ data points collected at altitude > 8 km have been identified as stratospheric, although this fraction varies with altitude, latitude and season (i.e. flight routes).

To calculate climatological distributions of $CO_2$ in the troposphere, we apply a method similar to Sweeney et al. (2015).

(1) The long-term trend of the flask-based $CO_2$ mole fraction data at Mauna Loa (MLO; 19.54°N, 155.58° W, 3397 m.a.s.l.), Hawaii, obtained from NOAA/ESRL/GMD (National Oceanic and Atmospheric Administration/Earth System Research Laboratory/Global Monitoring Division; available at ftp://aftp.cmdl.noaa.gov/data/) is calculated using a digital filtering technique (Nakazawa et al., 1997). The dataset goes to the end of 2015. In general, the long-term $CO_2$ trend at MLO is representative of the large-scale clean atmosphere and thus has been used as a reference site (Sweeney et al., 2015). (2)

Deviations of individual $CO_2$ data points from the long-term trend ($\Delta CO_2$) are calculated as

$$\Delta CO_2 \ (lat, lon, alt, t) = CO_2 \ (lat, lon, alt, t) - Trend \ CO_2 \ \text{at MLO} \ (t) \tag{1}$$

where *lat*, *lon*, *alt*, *t* are latitude, longitude, altitude and time of individual CONTRAIL CME data points, respectively, and *Trend $CO_2$ at MLO* is the long-term trend curve derived as described above. The CONTRAIL $CO_2$ data over 12 airports in Asia colorcoded by altitude are presented in Fig. 2, together with the MLO $CO_2$ data and the calculated long-term trend. In

this study, we present results from the statistical analysis of the $\Delta CO_2$ data (i.e. deviations of the individual data points from the black line in each panel of Fig. 2) for the years 2005–2015.



### 2.3 Model simulation

To better understand processes that generate the observed tropospheric distribution of $CO_2$ over the Asia-Pacific region, we analyze $CO_2$ simulated by the model NICAM-TM (Nonhydrostatic Icosahedral Atmospheric Model-based Transport Model) (Satoh et al., 2014). Details of the NICAM-TM $CO_2$ simulation and the evaluation of its performance have been

presented by Niwa et al. (2011, 2012). The atmospheric $CO_2$ transport is calculated using the 6-hourly meteorological data nudged to the JRA-55 reanalysis. The horizontal model grid interval is about 240 km and the number of vertical model layers is 40. For $CO_2$ simulation, fossil fuel (FF) emissions are obtained from the CDIAC (Carbon Dioxide Information Analysis Center) database (version 2013) (Andres et al., 2013), while fire emissions are from the GFED (Global Fire Emission Database version 3.1) (van der Werf et al., 2010). A priori terrestrial biospheric (BIO) fluxes are derived from the

CASA (Carnegie-Ames-Stanford Approach) model (Randerson et al., 1997). The air-sea exchange is based on the JMA (Japan Meteorological Agency) ocean flux data (Iida et al., 2015). The BIO fluxes are optimized in the NICAM-TM model inversion by using the GLOBALVIEW data (http://www.esrl.noaa.gov/gmd/ccgg/globalview/) and the CONTRAIL data in the FT (Niwa et al., 2012). Thus, the simulated atmospheric $CO_2$ is obtained from the optimized fluxes. We also examine simulated $CO_2$ fields driven by two different emission fluxes: one by FF and the other by BIO (hereafter referred as FF $CO_2$

and BIO $CO_2$, respectively). For comparison, the simulated data are sampled at times and locations coincident with the individual CONTRAIL CME data points, and processed in the same manner; stratospheric data are excluded by the model PV values; all $CO_2$ data points are detrended by the MLO long-term trend in the model.

### 3 Results

#### 3.1 Seasonal cycle of $CO_2$ in the UT over the Asia-Pacific region

Figure 3 presents monthly averaged distributions of the UT $\Delta CO_2$ over the Asia-Pacific region (left panels) along with histograms in the respective 5° latitude bands (right panels). In the left panels, the black arrows indicate monthly averaged horizontal wind at 250 hPa pressure surface from the JCDAS reanalysis. We note that monthly $CO_2$ distributions from the CONTRAIL data previously presented by Sawa et al. (2012) were calculated as averages in 20° (longitude) × 10° (latitude)

bins. In this study we were able to increase the spatial resolution to 5° × 5° since we have more data. As seen in Fig. 3, the UT $\Delta CO_2$ undergoes a clear seasonal cycle that varies significantly with latitude and longitude.

In January–February, the UT $\Delta CO_2$ is relatively uniform in space (Figs. 3a and 3c), except in regions > 35° N where the histograms show occurrences of higher $\Delta CO_2$ values (Figs. 3b and 3d). In March, high $\Delta CO_2$ values are apparent in regions > 30° N over northern Japan and downwind (Fig. 3e) where significantly increased frequency of high $\Delta CO_2$ up to 6 ppm are

observed (Fig. 3f). This feature becomes more pronounced in April (Fig. 3h) with expanded areas of high $\Delta CO_2$ around Japan (Fig. 3g). By May, regions with high $\Delta CO_2$ extend to > 20° N (Figs. 3i and 3j).



By June, the observed high $\Delta CO_2$ values over Japan and the northwestern Pacific nearly disappear (Fig. 3k). A significant fraction of the low $\Delta CO_2$ values down to −6 ppm and lower is observed at latitudes > 35° N (Fig. 3l). Due to these low $CO_2$ appearing at northern latitudes, the latitudinal gradient of the UT $\Delta CO_2$ starts to reverse (i.e. northward positive to negative) after June, aided by a moderately elevated $CO_2$ observed at 15°–30° N. In July, we begin to see very

low $\Delta CO_2$ values below −6 ppm in high latitude regions (> 40° N), particularly over the boreal Eurasia (Figs. 3m and 3n). To the south, only very small spatial gradients are observed.

In August, we see the $CO_2$ decrease broadly at all latitudes over the Asia-Pacific region, with a distinctly low $\Delta CO_2$ values forming over South Asia to Southeast Asia (Fig. 3o). The UT wind field shows anticyclonic wind circulation pattern over this region. This wind structure is coincident with the distinct low $\Delta CO_2$ observed over the continent, indicating that the

low $CO_2$ air mass is confined within the UT anticyclone. This clear $CO_2$ spatial structure associated with the anticyclone is for the first time depicted by the improved spatial resolution of the CONTRAIL data since Sawa et al. (2012). It is noted that such distinct low-$CO_2$ structure does not appear until July, despite the fact that the anticyclonic wind pattern starts in June (Figs. 3k and 3m).

Moving into September, we see a further decrease in $\Delta CO_2$ across the wider Asia-Pacific region (Figs. 3q and 3r). The

persistent UT anticyclonic structure is still observable in both $\Delta CO_2$ and wind fields, but the sharp boundary along the East Asian coast that was seen in August (i.e. longitudinal gradient or contrast between the continent and the ocean) is now to some degree blurred (see also Fig. 7a). In October, the anticyclonic low-$\Delta CO_2$ feature diminishes and $\Delta CO_2$ is now relatively uniform in the observation region. Thereafter $\Delta CO_2$ increases as a whole during the winter until the return of the spring.

**3.2 Vertical gradients of $CO_2$ over Asian cities**

Figure 4 presents a climatology of seasonal variations and vertical profiles of $\Delta CO_2$ over 12 airports in Asia, as uniquely obtained by the CONTRAIL observation. We consider these figures to represent large-scale (regional) features in the FT as a result of detrending and binning of the data (500-m and 14-day averages from multiple-year data). At lower altitudes, relatively local features can be visible due to BL processes and flight route biases near the airports, but examining

such smaller-scale phenomena in detail is out of scope in this study. We also calculate for each airport, altitude variation of the standard deviation (SD) of $\Delta CO_2$ using two-weekly values obtained at each altitude bin (Fig. 5), as an extended update of Shirai et al. (2012) who addressed synoptic-scale $CO_2$ variability over NRT. This type of analysis is made possible due to CONTRAIL's high-frequency measurements during ascent/descent over the airports.

Stephens et al. (2007) compiled $CO_2$ data from flask-based aircraft observations at 12 sites around the world for

comparison with model simulations. Flask $CO_2$ data at 16 sites from the NOAA/ESRL aircraft program were reported by Sweeney et al. (2015), including some of the data analyzed by Stephens et al. (2007). The measurements by Sweeney et al. (2015) have revealed climatological $CO_2$ variations over North America, whereas the present study focuses on Asia with more frequent in-flight observations. Since vertical profile measurements are relatively scarce over Asia (see supporting



online material by Stephens et al., 2007), the CONTRAIL observation provides a greater spatiotemporal insight into regional carbon cycling processes. One of the remarkable features found in vertical $CO_2$ profiles from other regions is the dramatic decrease in $CO_2$ toward the ground in the summer period at mid-continental sites of the northern hemisphere (see Fig. S3 of Stephens et al., 2007 and Fig. 5 of Sweeney et al., 2015). Below we show that the vertical $CO_2$ profiles and their seasonal

changes observed by CONTRAIL in Asia are interestingly different from those reported by the previous measurements in other regions.

      The seasonal $CO_2$ cycles with spring maxima and summer minima, typical for the northern hemispheric troposphere (Stephens et al., 2007; Sweeney et al., 2015), are to some degree obvious across regions over the 12 airports (Fig. 4) in Asia. However, a clear difference from those outside Asia is the general absence of a dramatic decrease of $CO_2$ near the ground in

the summer. In other words, the contoured low $\Delta CO_2$ in the summer is apparently "floating" in the FT and not connected to the ground, implying that the observed vertical profiles in the summer are not strongly influenced by uptake underneath. This feature is observed at all airports except DEL. In contrast, the springtime maximum $\Delta CO_2$ extends from the ground to the UT, indicating that the surrounding or upwind regions of most airports are strong sources of $CO_2$ during that season.

      NRT and HND, Japan, are the two airports over which the largest number of $CO_2$ measurements has been collected by

the CONTRAIL CME, giving relatively smooth climatology of $\Delta CO_2$ (Figs. 4b and 4c). Seasonal and vertical characteristics of $CO_2$ over HND and NRT are quite similar to each other. In the FT, $\Delta CO_2$ reaches its seasonal maximum in the spring (April–May) and minimum in the late summer to early autumn (September–October), with the seasonal amplitude in general decreasing with altitude. We also find substantially enhanced SD below ~2 km over HND and NRT in the winter (November–April) and summer (June–August) (Figs. 5b and 5c). The high summer variability propagates up to higher

altitudes (~6 km), presumably associated with enhanced vertical mixing in the summer. The vertical gradient in $CO_2$ is small (< 2 ppm) during the summer period (June–September), but a clear gradient is detectable for the rest of the year. These features are commonly observed over the other two Japanese airports Nagoya (NGO) (~260 km west of Tokyo) and Fukuoka (FUK) (~880 km west-southwest of Tokyo and ~850 km east-northeast of Shanghai). Also notable is that, in September, $CO_2$ decreases with altitude, this feature being observed widely over these four Japanese cities. $\Delta CO_2$ undergoes a seasonal cycle

with spring maximum and summer minimum also over ICN (~ 570 km northwest of FUK) (Fig. 4a), but the minimum occurs in late August to early September, about a month earlier than observed over the aforementioned Japanese airports. The low $\Delta CO_2$ in the BL is a characteristic that is not observed over Japan.

      Along the east coast of the continental East Asia, measurements are obtained over three cities: SHA, HKG and TPE (Figs. 4d, 4h and 4i, respectively). $\Delta CO_2$ increases from September until May when it reaches a seasonal maximum. The

seasonal minimum in the UT appears in September–October, lagging the LT minimum by about a month. We see remarkably high $\Delta CO_2$ values in the BL over SHA and HKG, these phenomena being particularly pronounced over SHA where we frequently observe $\Delta CO_2$ enhancements of > 20 ppm below 1 km. The elevated $\Delta CO_2$ in the winter season (November–April) is also characterized by high variability (Figs. 5d and 5h). Although the seasonal and vertical





characteristics of $CO_2$ over TPE appear to be essentially similar to those over SHA and HKG, our measurements are sparse during May–October.

ΔCO$_2$ over DEL shows a unique seasonal variation. We note that DEL is the only inland site, whereas the all other sites presented in this study are located near the coast. Prominent is the strong $CO_2$ drawdown throughout the troposphere in

August–September, with very little vertical gradient in the FT due to vigorous vertical mixing (Fig. 4g). Another interesting feature is the relatively low ΔCO$_2$ in the BL (< ~3 km) during January–March. This wintertime $CO_2$ stagnation over DEL was recently attributed to uptakes by crops (mainly wheat) grown in the winter season in the surrounding region (Umezawa et al., 2016).

Clear seasonal $CO_2$ variations are also visible over BKK (Fig. 4j). The seasonal maximum happens in March–April in

the LT and propagates upward. These 2 months correspond to a period of enhanced ΔCO$_2$ variability near the ground (Fig. 5j). Over SIN in the Southeast Asia, ΔCO$_2$ exhibits a measureable seasonal variation (Fig. 4k). The seasonal variation in the FT over SIN is similar in phase with that observed over BKK, but with comparatively reduced magnitude. It should be also noted that, over SIN, the vertical gradient of ΔCO$_2$ is small throughout the year. A maximum vertical ΔCO$_2$ difference is only ~2 ppm observed in the boreal spring. Lastly, for CGK in the tropical Asia (Fig. 4l), the observed seasonality in ΔCO$_2$

in the LT is hard to characterize due to relatively large variability. But interestingly, the seasonal phases are apparently different below and above 2.5 km. Below that height, relatively high ΔCO$_2$ values appear during August–October, while, over the same period, ΔCO$_2$ in the FT decreases until the October minimum.

### 3.3 Simulated $CO_2$ distributions in the UT over the Asia-Pacific region

In Fig. 6, simulated (second column) monthly $CO_2$ distributions in the UT are compared to the observation (first column). The model outputs are sampled at location and time coincident with the observation and analyzed in the same manner as the measurements. The third and fourth column panels show the simulated FF $CO_2$ and BIO $CO_2$, respectively. We do not present a contribution from biomass burning, since it is relatively minor (though not negligible) in evaluating the seasonal variation. Also shown are monthly $CO_2$ distributions at 250 hPa pressure surface for 2011 (last column). We have

chosen the model year 2011 as a representative year whose seasonal $CO_2$ distribution patterns are not exceptional, although the simulated $CO_2$ exhibits interannual variation due to year-by-year changes in meteorology and $CO_2$ fluxes. Note that the model data in the last column are simple monthly averages at model resolutions; thus, both the UT and stratospheric model data are included and avoids sampling bias that might result from data availability as in the observation. However, similarity in the features between the model and observed results, together with the model monthly averages, attest to the fact that the

CME-based $CO_2$ distribution is representative of the seasonal $CO_2$ climatology in the UT.

By comparing with the observation, we see that NICAM-TM (second column) is able to reproduce the overall general seasonal features of the observed $CO_2$ distribution pattern in the UT over the Asia-Pacific region. The model simulation (second column) shows seasonal $CO_2$ elevations centered at 20°–40° N in April–June, depletions of $CO_2$ over the boreal



Eurasia starting from June, and a distinct decrease in $CO_2$ over South Asia to Southeast Asia in August–September, all of which are in agreement with the observation (first column). In Section 4 below, we discuss how these features constitute the large-scale seasonal $CO_2$ distributions, as depicted in the last column. One notable feature that is not well reproduced by the model is the high $\Delta CO_2$ values observed over northern Japan in April, the cause of which is yet to be determined.

# 4 Discussion

## 4.1 Summertime $CO_2$ drawdown

In Section 3.1, we presented two major features in the $CO_2$ distribution in the UT over the Asia-Pacific region in the boreal summer: (1) the distinct low $CO_2$ values associated with the monsoon anticyclone over South Asia to Southeast Asia

during August–September and (2) the highly variable low $CO_2$ values at northern latitudes (> 40° N) during June–August (Fig. 3). These summertime low-$CO_2$ phenomena are hereafter referred to as the "monsoon low $CO_2$" and "boreal low $CO_2$", respectively.

### 4.1.1 Monsoon low $CO_2$

In August, a distinct circular-shaped distribution of low $CO_2$ over the Asian continent is prominent in both the observed and simulated $\Delta CO_2$ (Figs. 6u and 6v). The model reproduces the observation well in terms of the location of the spatially minimum $CO_2$ (i.e. the low-$CO_2$ over South Asia and northern Southeast Asia). Although the CONTRAIL data are not available over inland China (in particular over the Tibetan Plateau), the model simulation (Fig. 6y) offers a complete picture of the UT low $CO_2$ distribution associated with the monsoon anticyclone. Interestingly, the anticyclonic $CO_2$ pattern is

mainly composed of low BIO $CO_2$ (Fig. 6x). The region of lowered $CO_2$ in the monsoon anticyclone expands until September, as the confinement of the low $CO_2$ in the anticyclone becomes less distinct than in August as region-wide decrease of $CO_2$ occurs (Figs. 6z, 6aa and 6ad). The simulation indicates spreading of low BIO $CO_2$ to the northwestern Pacific from the anticyclone. In October, the UT $CO_2$ becomes nearly uniform again over the entire Asia-Pacific region (Figs. 6ae, 6af and 6ai).

As described above, the monsoon low $CO_2$ in the UT anticyclone is seasonally most distinct in August (Figs. 3 and 6), which is in fact coincident with the dynamical development of the summer monsoon anticyclone. Previous studies have shown that dynamical strengths of the monsoon anticyclone and convective activity reach their seasonal maxima in July–August (Randel and Park, 2006; Garny and Randel, 2013), and that, consequently, the confinement of the air mass within the UT anticyclone is seasonally strongest in August (Rauthe-Schöch et al., 2016). The CONTRAIL flights have only DEL

where vertical profiles inside the monsoon low $CO_2$ could be collected (see Fig. 3o). The DEL measurements clearly illustrate $CO_2$ well-mixed in the FT with pronounced decrease in the BL (Fig. 4g). This feature is consistent with the



interpretation that the neighbouring region of DEL (i.e. northwestern India) is part of the vertical conduit core that effectively transports surface flux signals upward to the upper tropospheric part of the summer monsoon anticyclone (Bergman et al., 2013). Our simulation shows that the BIO uptake in South Asia plays a dominant role in lowering the UT $CO_2$ (Fig. 6). In this connection, model studies have demonstrated that aircraft data within the anticyclone have a significant

impact in constraining surface $CO_2$ fluxes in South Asia (Patra et al., 2011; Niwa et al., 2012). In August, over other Asian cities, such as SIN, BKK, HKG and SHA, the summertime $\Delta CO_2$ values are not as low as those in the monsoon anticyclone (Fig. 3o), which means that these cities are outside the monsoon vertical conduit.

In September, the vertical profiles over DEL (i.e. the core of the monsoon low $CO_2$) retain vertically well-mixed low $CO_2$ as in August (Fig. 5g). This is indicative of strong BIO $CO_2$ uptake, as reflected in the optimized flux (see Fig. 5d of

Niwa et al., 2012). At the same time, we see a region-wide $CO_2$ decrease, as the August sharp $CO_2$ gradient at the edge of the UT anticyclone becomes blurred (Fig. 3q). This implies a broad propagation of the monsoon low $CO_2$ in the UT, as the anticyclonic confinement weakens (Garny and Randel, 2013; Rauthe-Schöch et al., 2016). The expansion of the monsoon low $CO_2$ in the UT (Fig. 3q) is reflected in the vertical $CO_2$ profiles over HKG and SHA where substantial $CO_2$ decreases are observed in the UT in September (i.e. the vertical gradients of $CO_2$ over both cities increase from August to September; see

Fig. 4). A similar, but less pronounced, feature is observed further downwind over cities in Japan (FUK, NGO, NRT and HND), as it is advected by strong westerly winds to the western Pacific in October. The decreasing $CO_2$ with altitude in the late summer is unique over the Asia-Pacific region where outflow from the monsoon low $CO_2$ in the UT is a significant contributing factor. The same process involving the Asian summer monsoon anticyclone can be invoked to explain the elevated methane ($CH_4$) values of South Asian origin observed in the UT over the western Pacific in the summer (Umezawa

et al., 2012).

Figure 7a compares seasonal variations of $\Delta CO_2$ in the UT over the South Asian continent (75°–100° E) and the western Pacific Ocean (130°–150° E) at latitudes 20°–30° N. Here we define longitudinal gradient as the difference in $\Delta CO_2$ between these two continental and oceanic areas (black line). The observed longitudinal gradient is nearly zero in July, increasing rapidly to 2.5 ppm in August, decreasing to 1.8 ppm in September, and then disappearing in October. This

seasonal change is reproduced well by the NICAM-TM simulation (Fig. 7b). We also show a break down of the longitudinal gradient into BIO and FF $CO_2$ contributions. Clearly, BIO $CO_2$ is the predominant driver of the seasonal $CO_2$ variation in the UT over both areas and contributes to the longitudinal gradient in August–September due to the monsoon anticyclone. Over South Asia, seasonal maximum contribution of BIO $CO_2$ to the summertime decrease is seen in August. This effect is not observed until September over the western Pacific, a lag on the order of a month.

It is interesting to note the difference in the timing between the $CO_2$ drawdown and the accumulation of other pollutants inside the UT monsoon anticyclone. As clearly seen in Fig. 3m, no enhancement/depletion in $CO_2$ is observed in the UT monsoon anticyclone in July. This is in contrast to studies that have indicated elevation of pollutant species within the monsoon anticyclone starting in June to July (e.g. Park et al., 2009; Xiong et al., 2009; Schuck et al., 2010; Randel et al., 2010). The difference between atmospheric $CO_2$ and other pollutants lies in the fact that these "other pollutants" are mostly



of anthropogenic origins that essentially have no seasonal cycle. The observed enhancement of these pollutants are therefore driven mostly by the anticyclone dynamics (Randel and Park, 2006; Park et al., 2009; Bergmann et al., 2013), and not by the seasonal variation in the surface emission, as in $CO_2$. Therefore, the absence of the anticyclonic structure in $CO_2$ in July is attributable to its surface flux characteristics. In July, the region's terrestrial biosphere might be still in transition from

overwhelming respiration (net source) to photosynthesis (net sink) (Niwa et al., 2012; Patra et al., 2013), since substantial precipitation arrives 1–2 months after the onset of the monsoon (i.e. prevailing southwest wind) in June (India Meteorological Department at http://imd.gov.in/pages/monsoon_main.php). From August to September, strong biospheric $CO_2$ uptake in South Asia takes place (Niwa et al., 2012), giving rise to the observed monsoon low $CO_2$ in the UT (Fig. 6) that is simulated well in our model.

### 4.1.2 Boreal low $CO_2$

In July, a prominent feature that is common in the CONTRAIL measurements and the NICAM-TM simulation (Figs. 6p and 6q) is a sharp north-south $CO_2$ gradient at 40°–50° N, with low values to the north and relatively uniform $CO_2$ to the south. In the NICAM-TM simulation, much of the BIO $CO_2$ uptake in the boreal Eurasia propagates to the northern Pacific

(Fig. 6s).

The boreal low $CO_2$ (i.e. the deeper drawdown of $CO_2$ and its earlier phase at higher latitudes) in the UT has been understood in the context of BIO $CO_2$ uptake propagating from mid to high latitudes (Tanaka et al., 1988; Nakazawa et al., 1991; Matsueda et al., 2002). It is estimated that a substantial $CO_2$ uptake by boreal biosphere starts in June and peaks in July to early August (e.g. Randerson et al., 1999; Saeki et al., 2013; Zhang et al., 2014), therefore the occurrence of the

boreal low $CO_2$ in the UT are consistent in phase with the atmospheric propagation of boreal BIO uptake. Sawa et al. (2012) showed that, in summer, convective uplift of surface low-$CO_2$ air lowers $CO_2$ in the FT at the NH mid to high latitudes. As clearly seen in Fig. 3, we observed the large $CO_2$ variability (the wide spreads of the histograms, see panels l, n and p) in the UT north of 40° N in June–August. This large $CO_2$ variability can be explained most likely by sporadic occurrences of convection over boreal Eurasia, as well as to a lesser extent by seasonally strongest and heterogeneous BIO $CO_2$ uptake;

such an example from the CONTRAIL measurement flights has been presented in Fig. 5 of Sawa et al. (2012). Miyazaki et al. (2008) pointed out that the boreal low $CO_2$ in the summer is isolated from the lower latitudes due to slow mean meridional circulation and weak cyclonic activity during the season. This can be seen in the CONTRAIL data (Fig. 6p, see also Fig. 6 of Sawa et al. 2012) and the NICAM-TM simulation (Fig. 6q). It is also noted that the spread of the histogram over the boreal Eurasia decreases in September (Fig. 3r), inferring that the convective activity and BIO $CO_2$ uptake over the

continent seasonally weakens and the UT resumes "background" $CO_2$ after the summer period of large fluctuations.

The extent to which the boreal low $CO_2$ is advected has significant impact on the observed seasonal cycles of $CO_2$ in the LT over East Asian cities. As described earlier, the seasonal $CO_2$ minimum over ICN occurs about a month earlier than over Japan at similar latitudes (Fig. 4). Based on the NICAM-TM model analysis of the ICN measurements (Niwa et al., 2017), it



is found that air masses observed in the LT over ICN in the summer are influenced by surface fluxes in the boreal Eurasia. As mentioned earlier, the boreal BIO $CO_2$ uptake peaks in July, relatively earlier than at mid latitudes. Accordingly, larger contributions of air masses from the north in the early summer would lower atmospheric $CO_2$, shifting earlier the occurrence of the seasonal $CO_2$ minimum at mid latitudes.

### 4.2 Seasonally elevated and highly variable $CO_2$ in spring

We have shown in Section 3 that seasonally elevated $CO_2$ is observed throughout the whole troposphere over the East Asia region in April–May (Figs. 3 and 4). This elevated $CO_2$ is accompanied by increased spread of $\Delta CO_2$ values in the UT north of 30° N in March–April (Figs. 3f and 3h). As shown by the NICAM-TM simulation, seasonally high $CO_2$ can be
explained mostly by the BIO emission fluxes, but a significant portion of the associated variability is due to enhanced synoptic-scale meteorological variability.

One of the most likely factors in meteorology is the active passage of eastward-tracking synoptic systems. In East Asia, cyclonic activity is most frequent in the spring (Chen et al., 1991; Adachi and Kimura, 2007). In association with the eastward moving springtime cyclonic activity, two major transport pathways have been suggested for pollutant outflow from
the continental East Asia to different tropospheric layers over the northwestern Pacific. (1) The first mechanism involves the advection of polluted BL air behind the cyclonic cold front as it moves eastward over the East Asian continent out to the Pacific (Liu et al., 2003; Sawa et al., 2007). Consequently, periodic passages of cyclones produce episodic variations of anthropogenic trace gases in the BL across the northwestern Pacific (Liu et al., 1997; Liang et al., 2004; Sawa et al., 2007; Tohjima et al., 2010, 2014). (2) The second mechanism involves frontal uplift of air in front of a moving cold front, in what
is called the warm conveyor belt. The uplift frequently takes place over South and Central China and the plume travels northeastward along the warm conveyor belt to the northwestern Pacific (Bey et al., 2001; Liu et al., 2003; Miyazaki et al., 2003; Liang et al., 2004). Convective uplift along a frontal zone over Central China and Southeast Asia also transports BL air to the FT (Miyazaki et al., 2003; Oshima et al., 2004). In addition to the above transport processes associated with cyclones, orographic forcing over South and Central China has also been observed to uplift the BL air to the FT (Liu et al.,
2003). Once in the FT, the plume can be easily exported to the Pacific by the midlatitude westerly winds (Bey et al., 2001; Liu et al., 2003).

In summary, periodic and episodic cyclonic uplifting of the BL air over the continental East Asia, with strong surface $CO_2$ emissions could explain the seasonal maximum level of $CO_2$ and increased variability in the FT in the spring (Fig 3). Using the CONTRAIL data, Shirai et al. (2012) also showed that the observed synoptic-scale variability of the FT $CO_2$ over
NRT increases in the spring, as air influenced by the continental East Asian $CO_2$ emissions is advected towards Japan.



**5 Concluding Remarks**

We have presented spatiotemporal variations of tropospheric $CO_2$ over the Asia-Pacific region observed uniquely by the CONTRAIL commercial airliner measurements. High-frequency in-flight $CO_2$ measurements by the CONTRAIL CME cover large part of the Asia-Pacific region and contribute to an enhanced characterization and understanding of the

climatological distribution of $CO_2$ over the region. Some of the highlights in this study are summarized as follows.

In summer, the region-wide low $CO_2$ across the Asia-Pacific region is primarily due to the $CO_2$ drawdowns in two distinct regions: the monsoon low $CO_2$ and the boreal low $CO_2$. The monsoon low $CO_2$ reflects South Asian biospheric $CO_2$ uptake and its propagation in the UT in association with the development and decay of the Asian summer monsoon anticyclone. This process contributes significantly to the observed horizontal and vertical variations in $CO_2$ over the Asia-

Pacific region. The monsoon outflow increases in September as the anticyclone decays, delivering low $CO_2$ (from South Asian biosphere) to the UT over the northwestern Pacific. In contrast, the boreal low $CO_2$ is driven by boreal terrestrial biospheric uptake. Heterogeneous spatial distributions of the biospheric flux, combined with the sporadic convective vertical transport over the Eurasian continent, cause seasonally large variability in the UT $CO_2$ at north of 40° N.

In spring, active passages of the eastward-tracking synoptic system sweep the continental East Asia and transport the

region's $CO_2$ emissions up to the UT, elevating atmospheric $CO_2$ over the northwestern Pacific. These synoptic systems also increase variability in $CO_2$. Given the high-density CONTRAIL measurements over Asia, and in particular around Japan, the CONTRAIL data provide promising opportunity for diagnosing detailed transport processes by midlatitude cyclones of $CO_2$ emitted from the continental East Asia.

The CONTRAIL commercial airliner measurements over the Asia-Pacific region can be exploited in constraining

emissions of various trace gases from East Asia and South Asia, particularly in the context of the role of the Asian summer monsoon. Also, given the unique spatiotemporal measurements along high altitude cruise and vertical flights, the CONTRAIL data can be used to evaluate emerging greenhouse gas data obtained by satellites.

**Acknowledgement**

We are grateful to engineers and staffs of the Japan Airlines, JAL Foundation and JAMCO Tokyo for supporting the CONTRAIL project. We also thank Keiichi Katsumata, Hisayo Sandanbata and Eri Matsuura (NIES) for technical support. We thank Ed Dlugokencky for the NOAA's flask-based $CO_2$ data at Mauna Loa. We acknowledge efforts by NICAM developers of Atmosphere and Ocean Research Institute of the University of Tokyo, Japan Agency for Marine-Earth Science and Technology, and RIKEN. We thank Kaz Higuchi (York University, Canada) for his comments to improve the

manuscript. The CONTRAIL observation was financially supported by the research fund by Global Environmental Research Coordination System and by Environment Research and Technology Development Funds (2-1401 and 2-1701) from Ministry of the Environment, Japan and Environmental Restoration and Conservation Agency. The CONTRAIL CME data




are posted on NOAA/ObsPak (http://www.esrl.noaa.gov/gmd/ccgg/obspack/) and available on the Global Environmental Database of the Center for Global Environmental Studies of NIES (doi.org/10.17595/20180208.001).

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



**Table 1: List of the major airports of the CONTRAIL $CO_2$ measurements in the Asia-Pacific region. Vertical profile data taken over neighbouring airports (listed with two airport codes) were merged for data analysis; note that the airport locations for the first airport code are shown throughout the manuscript. Numbers of vertical profiles are as of December 2015.**

| Airport code | City | Latitude | Longitude | Elevation (m) | Number of vertical profiles |
|---|---|---|---|---|---|
| ICN/GMP | Incheon | 37.469 | 126.450 | 7 | 206 |
| NRT | Narita | 35.764 | 140.392 | 43 | 7017 |
| HND | Haneda | 35.553 | 139.781 | 6 | 3656 |
| NGO | Nagoya | 34.858 | 136.805 | 5 | 911 |
| FUK | Fukuoka | 33.584 | 130.452 | 9 | 193 |
| SHA/PVG | Shanghai | 31.198 | 121.339 | 3 | 456 |
| DEL | Delhi | 28.566 | 77.103 | 237 | 715 |
| TPE/TSA | Taipei | 25.078 | 121.233 | 32 | 243 |
| HKG | Hong Kong | 22.309 | 113.915 | 6 | 662 |
| BKK | Bangkok | 13.681 | 100.747 | 2 | 1445 |
| SIN | Singapore | 1.350 | 103.994 | 7 | 838 |
| CGK | Jakarta | −6.126 | 106.656 | 10 | 407 |



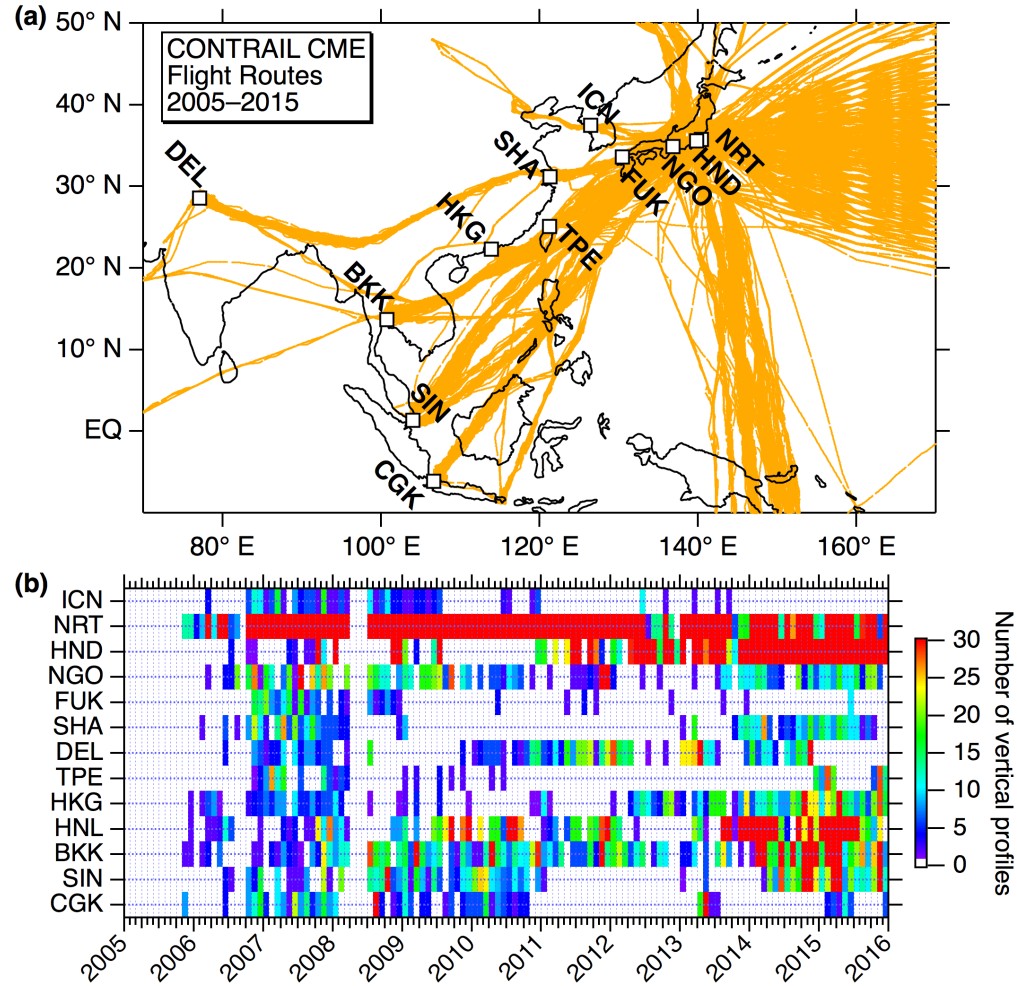

Figure 1: (a) A map showing flight tracks of the aircraft carrying the CME during 2005–2015. Airports highlighted in this study are shown by open squares with airport codes (Table 1). (b) Number of monthly vertical profiles taken over each airport. The airports are ordered north to south according to latitude (top to bottom).





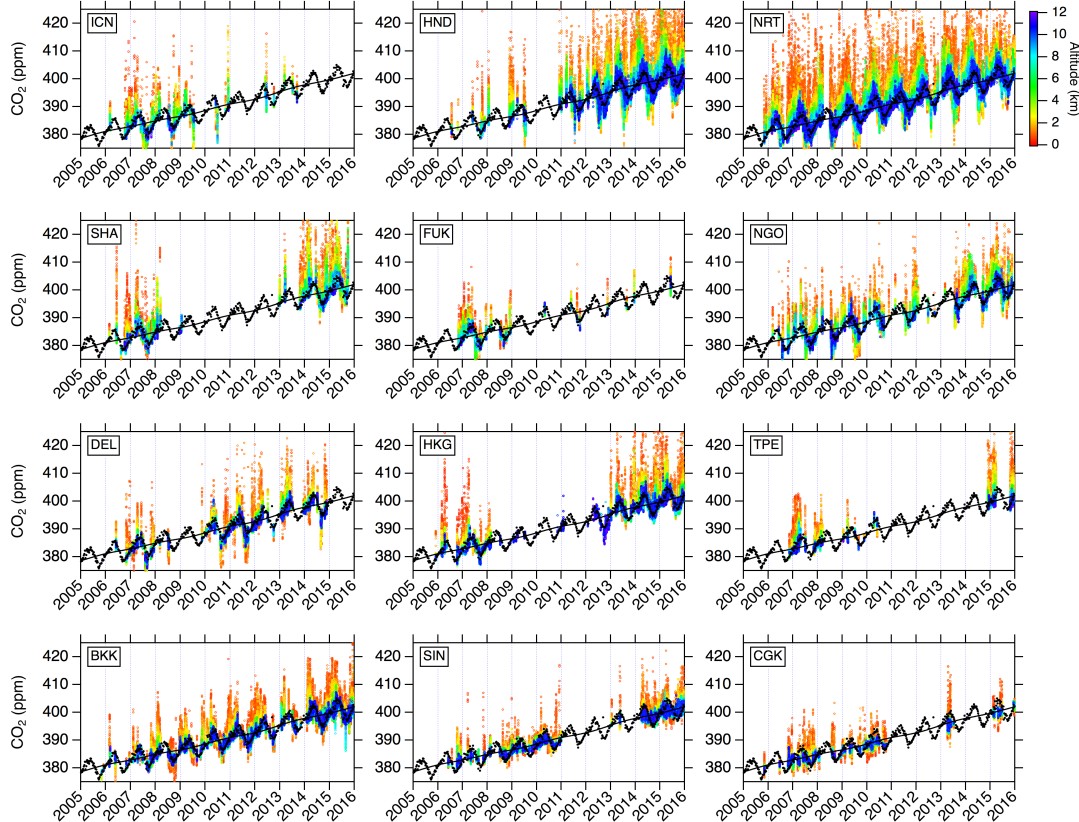

**Figure 2: Temporal variations of CO₂ over various airports in Asia. See Table 1 and Fig. 1 for the airport codes. Individual CO₂ data points are colored by altitude. The CO₂ data over the two Shanghai airports (SHA and PVG) are merged and designated as SHA, and same for ICN (ICN and GMP) and TPE (TPE and TSA). Also shown in each panel for comparison are the flask-based**
5 **CO₂ data (black circles) and the long-term trend (black line) at the Mauna Loa Observatory (MLO; 19.54° N, 155.58° W, 3397 m above sea level; data obtained from the NOAA/ESRL/GMD).**




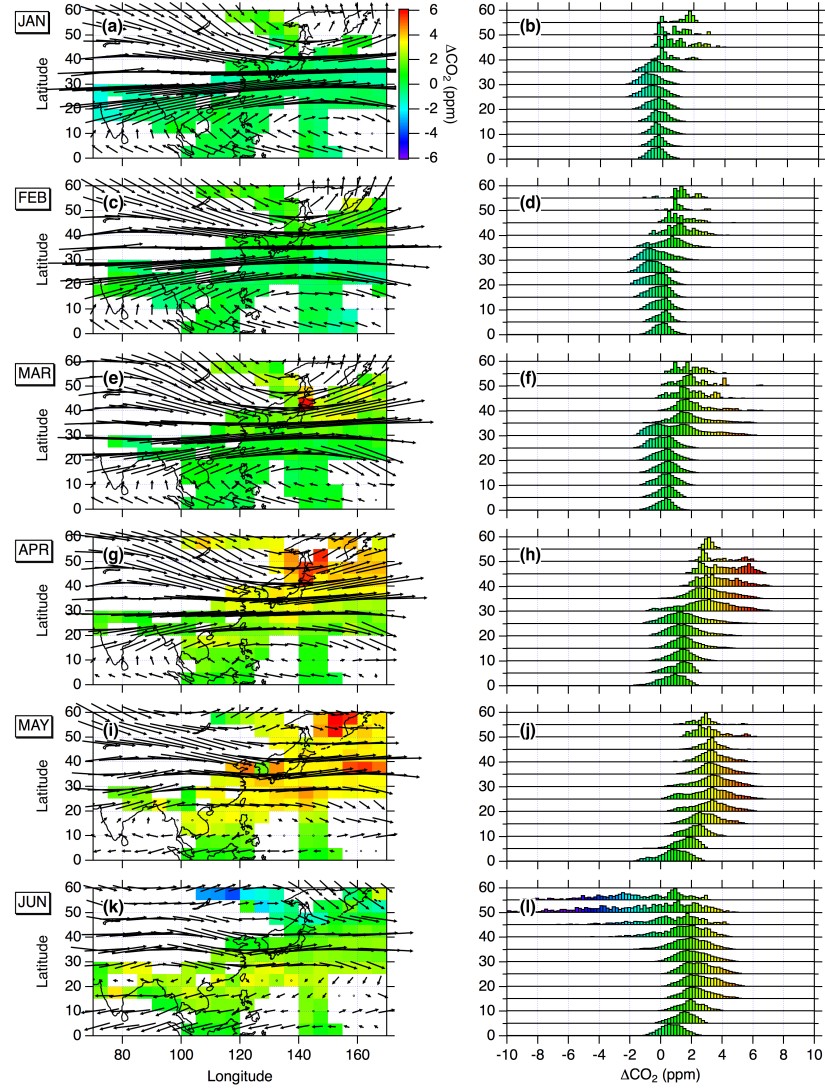

**Figure 3:** (Left) Monthly climatological CO$_2$ mole fraction ($\Delta$CO$_2$) in the UT over the Asia-Pacific region. The CO$_2$ data taken at altitudes > 8 km are averaged in each 5°×5° bin. The CO$_2$ data influenced by stratospheric air (PV > 2 PVU) were excluded. Also shown are monthly averaged wind vectors at 250 hPa. (Right) Histograms of $\Delta$CO$_2$ in each 5° latitude bands colorcoded in the same manner as in the left panels. Every histogram is normalized by maximum frequency.





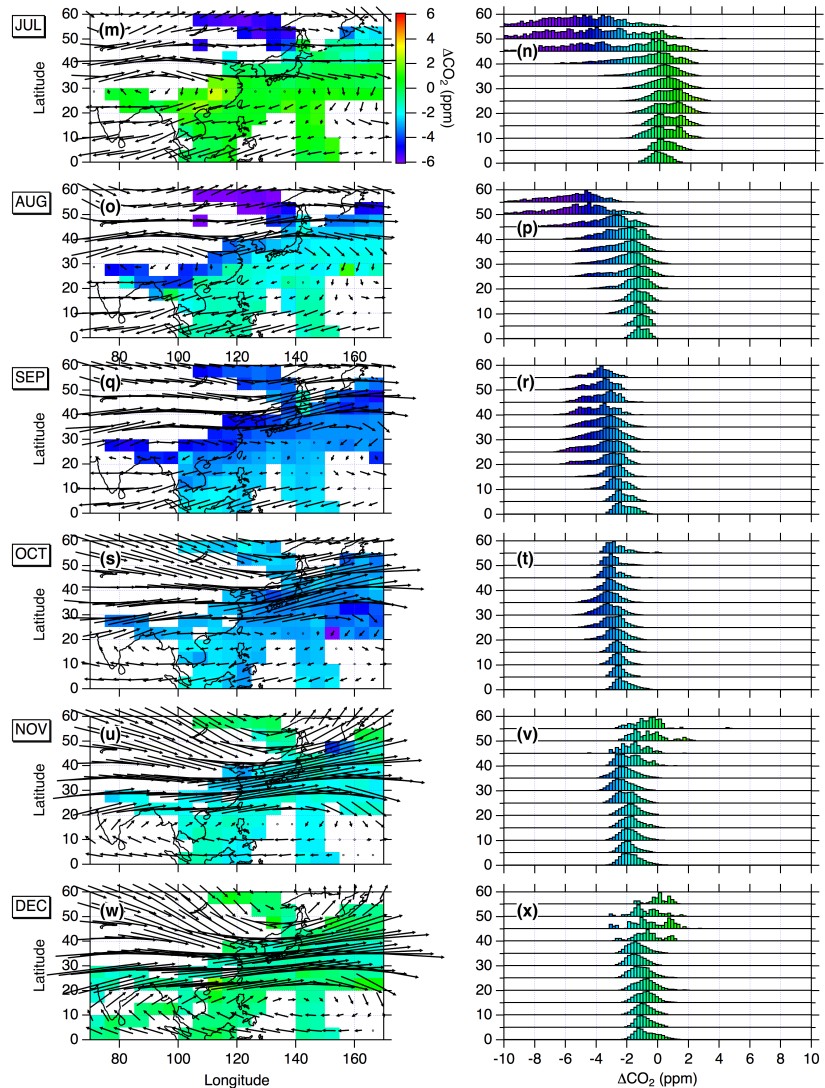

**Figure 3: (continued).**



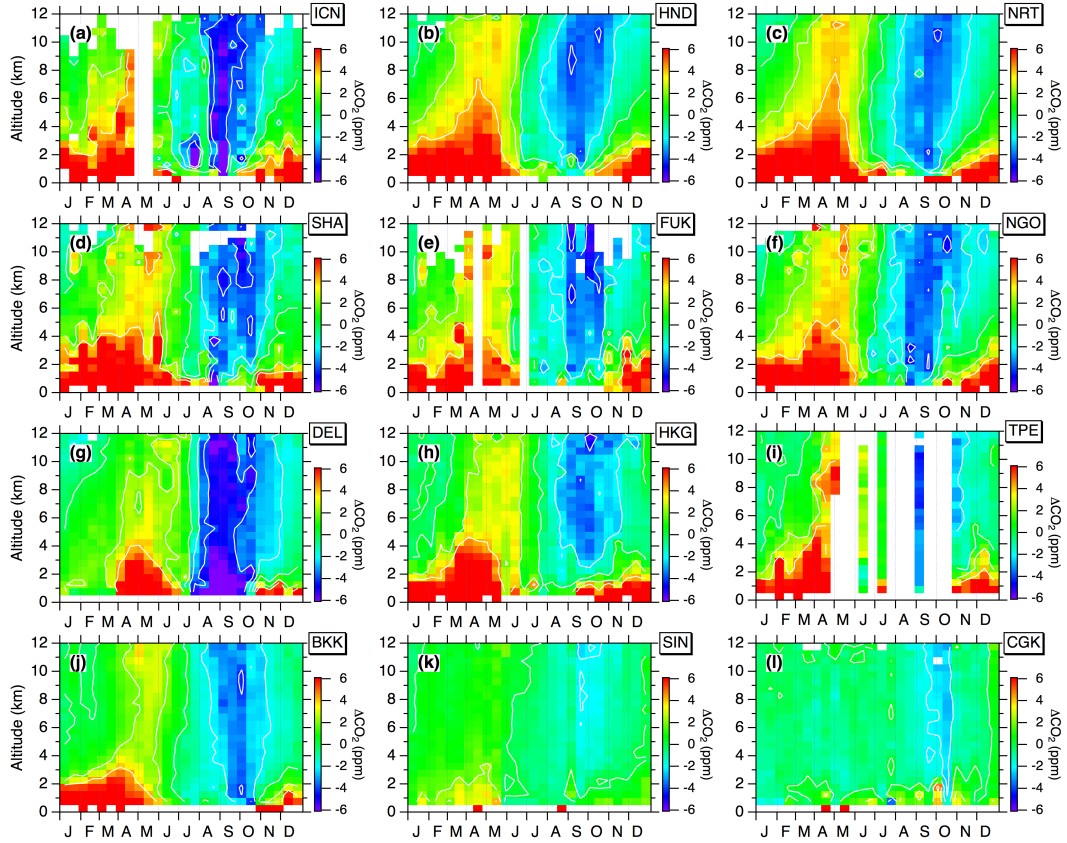

**Figure 4: Seasonal variations of vertical profiles of ΔCO₂ over (a) ICN, (b) HND, (c) NRT, (d) SHA, (e) FUK, (f) NGO, (g) DEL, (h) HKG, (i) TPE, (j) BKK, (k) SIN, and (l) CGK. The airport codes are listed in Table 1. The CO₂ data over some airports are merged, as described in Fig. 2. Vertical and horizontal bins are 500-m and 14-day intervals, respectively. White lines indicate isolines of ΔCO₂.**





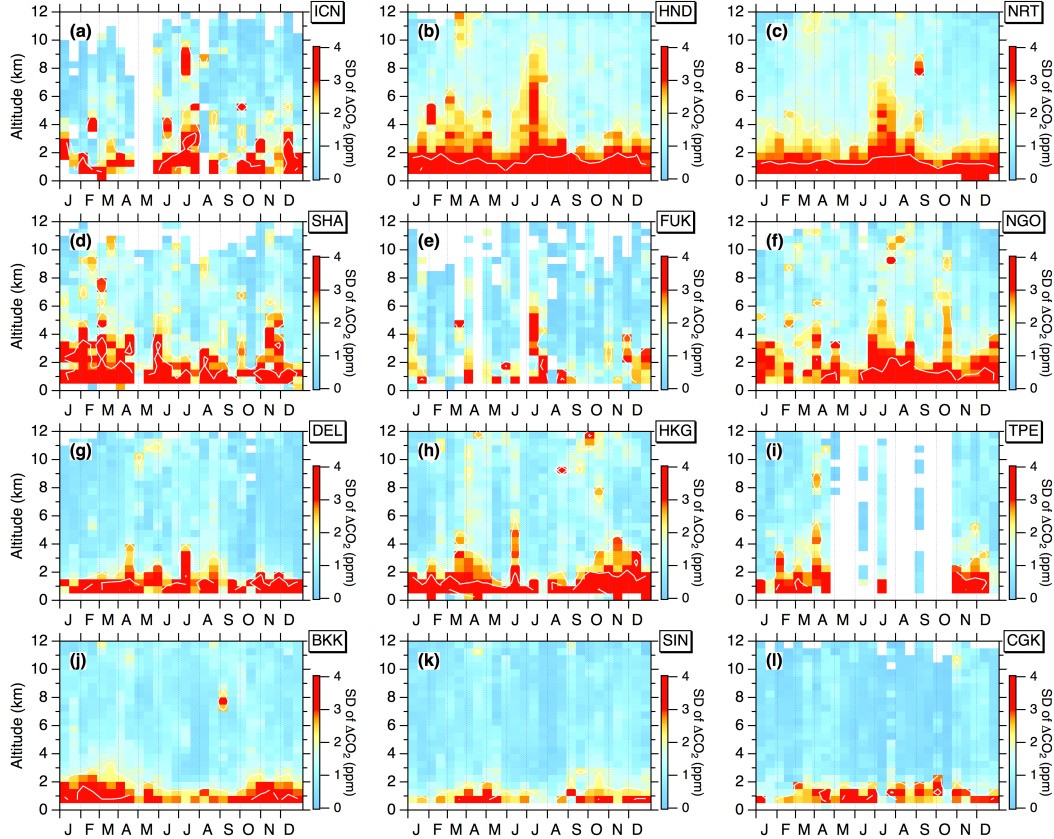

**Figure 5: Same as in Fig. 4 but for standard deviations of $\Delta CO_2$ in each bin. The standard deviation is calculated only when the bin has > 5 data points.**





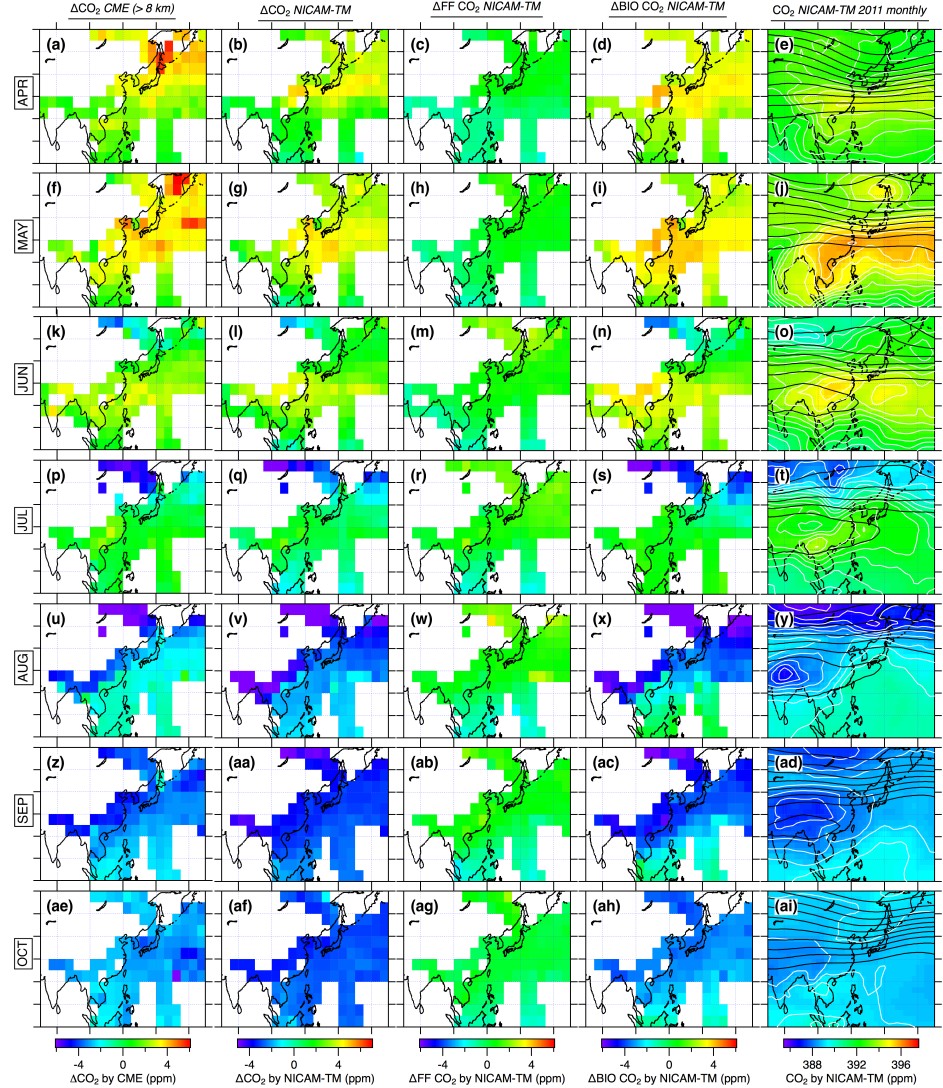

**Figure 6:** Climatological distributions of (first column) $CO_2$ in the UT (> 8 km) observed by CONTRAIL CME in comparison to (second column) $CO_2$, (third column) FF $CO_2$, and (fourth column) BIO $CO_2$ simulated by NICAM-TM. The CONTRAIL data are simply averaged for each grid, and the NICAM-TM data are sampled at locations and times corresponding to the observation data and analyzed in the same manner. Also shown are the simulated monthly distributions of $CO_2$ at 250 hPa pressure surface in 2011 (last column). Solid lines in white and black in the last-column panels indicate $CO_2$ and geopotential height at 250 hPa pressure surface, respectively.





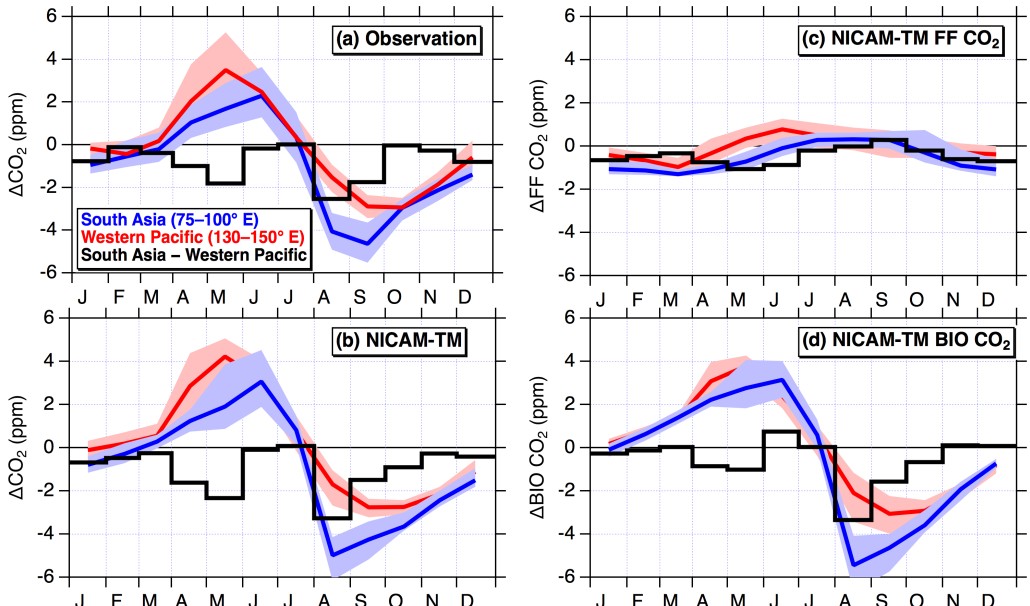

**Figure 7. (a)** Seasonal variations of $\Delta CO_2$ in the UT over South Asia (blue, 20°–30° N, 75°–100° E) and Western Pacific (red, 20°–30° N, 130°–150° E). Lines and shades are monthly medians and 25 and 75 percentiles, respectively. Black solid line shows monthly difference of $\Delta CO_2$ between the two areas (South Asia – Western Pacific i.e. longitudinal gradient). **(b)** Same as in (a), but for the NICAM-TM simulated data. **(c)** Same as in (b), but for FF $CO_2$ in the model. **(d)** Same as in (b), but for BIO $CO_2$ in the model.