# Peer review of "Seasonal evaluation of tropospheric CO2 over the Asia-Pacific region observed by the CONTRAIL commercial airliner measurements"

_Atmospheric Chemistry and Physics, 2018_

## Referee Comment (RC1) · Anonymous Referee #1 · 23 Jul 2018

General comments: This paper reports 10 years of CO2 measurements from the upper troposphere and from vertical profiles above 16 airports across Asia, obtained from commercial airline flights by the CONTRAIL program. This data set is extensive, high quality, unique and especially valuable for the reason that it defines the CO2 field above the surface in a sparsely observed part of the atmosphere.

In this study the authors investigate the upper tropospheric CO2 distribution over the Asia-Pacific region. They focus on some notable features, for example zones of low summertime CO2 above East Asia and boreal Asia, and interpret them in terms of surface exchange and transport processes using the NICAM-TM model.

[Figure]

Specific comments: This is a good paper that makes a solid contribution to its field. My only issue with the science presented relates to the discussion of vertical profiles above Asian cities in section 3.2 (3rd paragraph). The claim is made that these profiles differ from others outside of Asia with "absence of a dramatic decrease of CO2 near the ground in the summer. . . . . . . .implying that the observed vertical profiles in the summer are not strongly influenced by uptake underneath". My concern is that by nature of this program where the vertical profiles are above large population centres (and CO2 source regions), there may be a bias towards higher CO2 in the boundary layer than what was observed in vertical profile data elsewhere. The authors should address this possibility.

There is one section where some clarification and more detail is required. The 1st paragraph of section 2.1 (line 16-19 on page 3) describes standard gas measurement intervals. Where it is stated "intervals were initially 10 min. . . . .20 min..." it is not clear if the 10 and 20 minute intervals etc. refer to the duration of, or the time between standard analyses. It would be helpful to specify exactly what the analysis time cycles are. For example, during the 14 minute cycle, sample air is measured for x minutes, then standard 1 for y minutes and standard 2 for z minutes. It would also be useful to record what time or fraction of these data are rejected after switching gas streams.

Technical comments: A list of technical corrections follows. Many of these address overuse of "the" or "a". While the English used in the paper is generally very good, the readability could be easily improved by attention to these instances. Page 1, line 17 – delete "the" to leave "Pacific Rim of continental East Asia" P2, line 5 – "an increasing number" P2, line 16 – reword to ". . .less-well studied features of the CO2 distribution that are associated with the Asian monsoon." P2, line 31 – "another zone of low CO2" P3, line 27 – "flights to continental East Asia" P3, line 29 – "over continental Asia" P4, line 5 – reword to "Although measurements at other airports are less regular, data from sites where a substantial number of vertical profiles have been taken and cover much of the year, are included in this study." P4, section 2.2, 1st paragraph – It would be

appropriate to define here what is meant by upper troposphere. Figure 3 suggests altitudes > 8 km. It might also be worth briefly commenting on the upper boundary, presumably the tropopause, and how its height varies with latitude. P4, line 11 – "in atmospheric composition" P6, line 3 – "low CO2 values" and "of UT" P6, line 4 – "by moderately" P6, line 5 – "over boreal" P6, line 7 – "with distinctly" P6, line 22 – "by CONTRAIL observations" P6, line 23 – "500-m altitude" P6, line 24 – "due to boundary layer (BL) processes" P6, line 25 – ""is beyond the scope of this study" P7, line 1 – "CONTRAIL observations provide greater" P7, line 28 – "east coast of continental" P7, line 30 – "lagging the lower troposphere (LT) minimum" P8, line 7 – "uptake by crops" P8, line 11 – "measurable" P8, line 14 – "in tropical Asia" P8, line 20,28,31 – "observations" P8, line 33 – "depletion of CO2 over boreal" P10, line 9 – Fig. 4g instead of 5g P11, line 14 – "in boreal" P11, line 20 – "in the UT is consistent" P11, line 29 – "over boreal", also replace "inferring" with "implying" P12, line 1 – "in boreal" P12, line 2 – delete "relatively" P13, line 14 – "sweep continental" P13, line 18 – "from continental" P13, line 21 – replace "flights" with "profiles" P16, line 4 – Matsueda and Inoue (1999) appears in the reference list but is not referred to in the text Figure 6 – 1) add y-axis (latitude) labels, 2) the black lines showing geopotential height in the last column are meaningless without some numerical labelling Figure 6 caption, lines 1-2 – columns 1 – 4 show ∆CO2, ∆FF CO2 and ∆BB CO2 Figure 6 caption, line 6 – CO2 isolines

---

## Referee Comment (RC2) · Anonymous Referee #2 · 1 Aug 2018

General comments:

This paper addresses the long-term tropospheric distributions of CO2 over the Asia-Pacific region obtained from the commercial airliner measurements under CONTRAIL project. High quality tropospheric CO2 data in general are sparse and such data specially the rapid developing Asian regions are specially limited. These long-term observations can contribute to constrain the emission patterns for the rapid developing Asian region that is critically important to the global carbon budget. The text provides a good summary of upper tropospheric CO2 distributions and role of the responsible factors for the seasonal distribution over Asia-Pacific region. I acknowledge the large amount

of work provided by the authors and interesting information issued from this study. This work is interesting to be published and is fully within scope of ACP

Technical Comments:

Abstract: Please include 2-3 sentences for highlighting the importance of the study.

Abstract: Line 18: "It is found. . .. season" – The sentence is long and not clear to me. Please reformulate it.

Introduction: Line 27: "China is now . . ..nations" – The sentence is not clear. Please reformulate it.

Figure1: Please tag the climatological mean $CO_2$ concentrations along with the flight tracks in "a" panel if possible.

Figure 3. Please mentioned the source of wind vector data in the caption.

Figure 3 and 4. Please remove the wind vectors at the boundaries of each boxes. Also reduce the wind vector density. The anticyclonic feature from the wind vectors is not very clearly. The author could try to plot the wind vectors at 215 hPa or 200 hPa for better visualization of anticyclone if possible. The following study can be refer for example

Park, M., W. J. Randel, L. K. Emmons, and N. J. Livesey (2009), Transport pathways of carbon monoxide in the Asian summer monsoon diagnosed from Model of Ozone and Related Tracers (MOZART), J. Geophys. Res., 114, D08303, doi:10.1029/2008JD010

Chandra, N., Hayashida, S., Saeki, T., and Patra, P. K.: What controls the seasonal cycle of columnar methane observed by GOSAT over different regions in India?, Atmos. Chem. Phys., 17, 12633-12643, https://doi.org/10.5194/acp-17-12633-2017, 2017.

Figure 3 and 4. The histogram panel looks too messy. The author can consider 10o latitudinal band instead of 5o for plotting histogram.

---

## Author Comment (AC1) · 12 Sep 2018

**Response to Referee #1**

We thank the referee for helpful comments to improve this paper. Our responses are detailed below. Please note that **referee's comments** and our responses are in different styles.

**General comments: This paper reports 10 years of CO2 measurements from the upper troposphere and from vertical profiles above 16 airports across Asia, obtained from commercial airline flights by the CONTRAIL program. This data set is extensive, high quality, unique and especially valuable for the reason that it defines the CO2 field above the surface in a sparsely observed part of the atmosphere.**

**In this study the authors investigate the upper tropospheric CO2 distribution over the Asia-Pacific region. They focus on some notable features, for example zones of low summertime CO2 above East Asia and boreal Asia, and interpret them in terms of surface exchange and transport processes using the NICAM-TM model.**

**Specific comments: This is a good paper that makes a solid contribution to its field. My only issue with the science presented relates to the discussion of vertical profiles above Asian cities in section 3.2 (3rd paragraph). The claim is made that these profiles differ from others outside of Asia with "absence of a dramatic decrease of CO2 near the ground in the summer. . .. . ...implying that the observed vertical profiles in the summer are not strongly influenced by uptake underneath". My concern is that by nature of this program where the vertical profiles are above large population centres (and CO2 source regions), there may be a bias towards higher CO2 in the boundary layer than what was observed in vertical profile data elsewhere. The authors should address this possibility.**

We thank the referee for recognizing the value of our data and for the important suggestion. It is true that our observations have collected vertical profiles from/to airports adjacent to big cities, and that the measurements, especially in the BL, are subject to influence from nearby urban emissions. We are aware of this feature and indeed have found notable increases of $CO_2$ in the BL over some airports. For instance, histograms of $\Delta CO_2$ in the BL over the Asian airports show a distribution with an

extended tail toward positive values than a compact Gaussian distribution. We could therefore have redrawn Figure 4 with median values, instead of averages, to reduce the effect of "polluted" profiles. However, the difference between average and median below 2 km falls mostly within < 1 ppm throughout the year over all the airports in Figure 4, except SHA and HKG where the value is ~1.5 ppm on yearly average. In fact, visual difference between average- and median-based Figure 4 is small. We note that the occurrence of "polluted" profiles is dependent on several factors, such as airport location relative to a nearby city, magnitude of nearby emissions and local meteorology. These features will be addressed in our future publication. In summary, although an "airport bias" likely has significant contribution in the BL over some of the CONTRAIL airports, we consider that the effect is small within the scope of this study. The following new paragraph has been added in the section for clarification on the issue:

"It is likely that some features shown in Fig. 4, especially in the BL, are due to the influence of nearby $CO_2$ emissions. Indeed, at some airports, large elevation of $CO_2$ values have been observed frequently in the BL. In order to reduce possible bias due to such pollution events, we did redraw Figure 4 with median $\Delta CO_2$ values, instead of averaged values. We found no clear visual difference in the overall features discussed below. In fact, differences between average and median are mostly < 1 ppm even below 2 km at all airports, except SHA and HKG where the value is ~1.5 ppm on yearly average. Although pollution events are observed frequently over these two airports (as described below), we consider such "airport bias" in the climatological vertical profiles to be small within the scope of this study. Influence of nearby city emissions on the CONTRAIL observations will be addressed in our future publication."

In addition, considering that vertical profile of $CO_2$ is determined by balance between uptake and emission in footprint areas, the original sentence has been modified to:

"…, implying that the observed vertical profiles in the summer are not dictated by overwhelming uptake underneath."

*There is one section where some clarification and more detail is required. The 1st paragraph of section 2.1 (line 16-19 on page 3) describes standard gas measurement intervals. Where it is stated "intervals were initially 10 min. . ...20 min..." it is not clear if the 10 and 20 minute intervals etc. refer to the duration of,*

*or the time between standard analyses. It would be helpful to specify exactly what the analysis time cycles are. For example, during the 14 minute cycle, sample air is measured for x minutes, then standard 1 for y minutes and standard 2 for z minutes. It would also be useful to record what time or fraction of these data are rejected after switching gas streams.*

We have now added the following sentences (underlined):

"The standard gases are currently introduced into the NDIR cell every 14 min during the ascent/descent portion of the flight and every 62 min during the constant altitude portion of the flight (cruise) typically at 8–12 km i.e. during the ascent/descent (cruise) measurement cycle, sample air is measured for 12 (60) minutes, then standards 1 and 2 are measured for 1 minute each. These standard gas intervals were initially 10 min during ascent/descent and 20 min during cruise until December 2005; the 20-min interval was then changed to 40 min until October–November 2011. The CME data are recorded as 10-s averaged measurements during ascent/descent (~100-m intervals in altitude) and at 1-min intervals during cruise (~15 km intervals horizontally). The data are rejected for 40 s after switching the gas stream and also when a standard deviation for the average period exceeds 3 ppm and when any failure in pressure/flow control is observed in the CME data record."

*Technical comments: A list of technical corrections follows. Many of these address overuse of "the" or "a". While the English used in the paper is generally very good, the readability could be easily improved by attention to these instances.*

We thank the referee for many technical corrections. Our responses follow below.

*Page 1, line 17 – delete "the" to leave "Pacific Rim of continental East Asia"*

Corrected.

*P2, line 5 – "an increasing number"*

Corrected.

*P2, line 16 – reword to ". . .less-well studied features of the CO2 distribution that are associated with the Asian monsoon."*

Corrected.

**P2, line 31 – "another zone of low CO2"**

Corrected.

**P3, line 27 – "flights to continental East Asia"**

Corrected.

**P3, line 29 – "over continental Asia"**

Corrected.

**P4, line 5 – reword to "Although measurements at other airports are less regular, data from sites where a substantial number of vertical profiles have been taken and cover much of the year, are included in this study."**

Corrected as follows:

"Although measurements over other airports are less regular, data from sites where a substantial number of vertical profiles have been taken and cover much of the year are included in this study."

**P4, section 2.2, 1st paragraph – It would be appropriate to define here what is meant by upper troposphere.**

The following sentence has been added at the end of the paragraph:

"In this study, the UT is defined as the region at altitudes of > 8 km and with PV of < 2 PVU."

**Figure 3 suggests altitudes > 8 km. It might also be worth briefly commenting on the upper boundary, presumably the tropopause, and how its height varies with latitude.**

This issue has been addressed in our previous papers (Sawa et al. 2008, 2012, 2015). We have however added the following sentence:

"Note that most commercial airliners cruise at altitudes of 9–12 km, and that this cruising altitude region is deemed in large part stratospheric at higher latitudes (e.g. 86% and 64% of the data taken at > 40° N was stratospheric in January and July,

respectively), whereas it mostly resides in the UT at lower latitudes (< 10% of the data at < 30° N was stratospheric throughout the year)."

**P4, line 11 – "in atmospheric composition"**
Corrected.

**P6, line 3 – "low CO2 values" and "of UT"**
Corrected.

**P6, line 4 – "by moderately"**
Corrected.

**P6, line 5 – "over boreal"**
Corrected.

**P6, line 7 – "with distinctly"**
Corrected.

**P6, line 22 – "by CONTRAIL observations"**
Corrected.

**P6, line 23 – "500-m altitude"**
Corrected.

**P6, line 24 – "due to boundary layer (BL) processes"**
Corrected. We apologize for this being missed in the previous manuscript.

**P6, line 25 – ""is beyond the scope of this study"**
Corrected.

**P7, line 1 – "CONTRAIL observations provide greater"**
Corrected.

**P7, line 28 – "east coast of continental"**

Corrected.

**P7, line 30 – "lagging the lower troposphere (LT) minimum"**

Corrected. We apologize for this being missed in the previous manuscript.

**P8, line 7 – "uptake by crops"**

Corrected.

**P8, line 11 – "measurable"**

Corrected.

**P8, line 14 – "in tropical Asia"**

Corrected.

**P8, line 20,28,31 – "observations"**

Corrected.

**P8, line 33 – "depletion of CO2 over boreal"**

Corrected.

**P10, line 9 – Fig. 4g instead of 5g**

Corrected. We apologize for this mistake.

**P11, line 14 – "in boreal"**

Corrected.

**P11, line 20 – "in the UT is consistent"**

Corrected.

**P11, line 29 – "over boreal", also replace "inferring" with "implying"**

Corrected.

**P12, line 1 – "in boreal"**

Corrected.

**P12, line 2 – delete "relatively"**

Corrected.

**P13, line 14 – "sweep continental"**

Corrected.

**P13, line 18 – "from continental"**

Corrected.

**P13, line 21 – replace "flights" with "profiles"**

Corrected.

**P16, line 4 – Matsueda and Inoue (1999) appears in the reference list but is not referred to in the text**

We thank the referee for noting this omission in the paper. Since this reference is important in the history of the CONTRAIL observations and of interpretation of the data taken in the UT over the western Pacific, it is now refereed to in section 4.1.2.

**Figure 6 – 1) add y-axis (latitude) labels, 2) the black lines showing geopotential height in the last column are meaningless without some numerical labeling**

Both x- and y-axes labels have been added. We have also labeled the black contours of geopotential height.

**Figure 6 caption, lines 1-2 – columns 1 – 4 show ΔCO2, ΔFF CO2 and ΔBB $CO_2$**
**Figure 6 caption, line 6 – CO2 isolines**

The caption has been changed as follows:

"Figure 6: Comparison of the observed and simulated distributions of $CO_2$ in the UT. Column 1 shows $\Delta CO_2$ observed by CONTRAIL CME. Columns 2–4 show $\Delta CO_2$, $\Delta FF$ $CO_2$, and $\Delta BIO$ $CO_2$ simulated by NICAM-TM. The CONTRAIL data are simply averaged for each grid, and the NICAM-TM data are sampled at locations and times

corresponding to the observation data and analyzed in the same manner. Also shown are the simulated monthly distributions of $CO_2$ at 250 hPa pressure surface in 2011 (column 5). Solid lines in white and black in the column 5 indicate $CO_2$ isolines and geopotential height at 250 hPa pressure surface, respectively."

---

## Author Comment (AC2) · 12 Sep 2018

**Response to Referee #2**

We thank the referee for helpful comments to improve this paper. Our responses are detailed below. Please note that **referee's comments** and our responses are in different styles.

*General comments:*
*This paper addresses the long-term tropospheric distributions of CO2 over the Asia-Pacific region obtained from the commercial airliner measurements under CONTRAIL project. High quality tropospheric CO2 data in general are sparse and such data specially the rapid developing Asian regions are specially limited. These long-term observations can contribute to constrain the emission patterns for the rapid developing Asian region that is critically important to the global carbon budget. The text provides a good summary of upper tropospheric CO2 distributions and role of the responsible factors for the seasonal distribution over Asia-Pacific region. I acknowledge the large amount of work provided by the authors and interesting information issued from this study. This work is interesting to be published and is fully within scope of ACP.*

We thank the referee for recognizing the value of this work and helpful comments.

*Technical Comments:*
*Abstract: Please include 2-3 sentences for highlighting the importance of the study.*

We have added the following sentences at the beginning of the abstract:
"Measurement of atmospheric carbon dioxide ($CO_2$) is indispensable for top-down estimation of surface $CO_2$ sources/sinks by an atmospheric transport model. Despite the growing importance of Asia in the global carbon budget, the region has been monitored for atmospheric $CO_2$ only sparsely and our understanding of atmospheric $CO_2$ variations in the region (and thereby that of the regional carbon budget) is still limited. In this study, we present…"

*Abstract: Line 18: "It is found. . .. season" – The sentence is long and not clear to me. Please reformulate it.*

The sentence has been reformulated as follows:

"It is inferred that a substantial contribution to the UT $CO_2$ over the northwestern Pacific comes from the continental East Asian emissions in the spring, but in the summer monsoon season, the prominent air mass origin switches to South Asia and/or Southeast Asia with distinct imprint of the biospheric $CO_2$ uptake."

***Introduction: Line 27: "China is now . . ..nations" – The sentence is not clear. Please reformulate it.***

The sentence has been changed to:

"China is now the world's largest $CO_2$ emitter, and India, Japan, and the Republic of Korea are all in the world's top 10 emitting nations (Boden et al., 2016)."

***Figure1: Please tag the climatological mean CO2 concentrations along with the flight tracks in "a" panel if possible.***

According to the suggestion, we have added the annual average $\Delta CO_2$ field to panel a. The caption text has been changed accordingly.

***Figure 3. Please mentioned the source of wind vector data in the caption.***

We have added the following description (underlined):

"… Also shown are monthly averaged wind vectors at 250 hPa from the JCDAS/JRA-55 reanalysis data (averaged for the observation years)."

***Figure 3 and 4. Please remove the wind vectors at the boundaries of each boxes. Also reduce the wind vector density. The anticyclonic feature from the wind vectors is not very clearly. The author could try to plot the wind vectors at 215 hPa or 200 hPa for better visualization of anticyclone if possible. The following study can be refer for example***
***Park, M., W. J. Randel, L. K. Emmons, and N. J. Livesey (2009), Transport pathways of carbon monoxide in the Asian summer monsoon diagnosed from Model of Ozone and Related Tracers (MOZART), J. Geophys. Res., 114, D08303, doi:10.1029/2008JD010***
***Chandra, N., Hayashida, S., Saeki, T., and Patra, P. K.: What controls the seasonal cycle of columnar methane observed by GOSAT over different regions***

*in India?, Atmos. Chem. Phys., 17, 12633-12643,*

*https://doi.org/10.5194/acp-17-12633-2017, 2017.*

We thank the referee for the suggestion. The wind vectors at the boundary have been removed and the number of wind vectors has been now reduced. As in previous studies, including the studies suggested by the referee, the altitudinal center of the anticyclone is higher than the typical cruising altitudes of commercial airliners. We therefore agree that the anticyclonic feature would be better visualized if we plotted the wind vectors at the pressure surfaces of ~200 hPa. However, in this study, it has been our intention to show that our observations by commercial airliners can scan part of the anticyclone and its $CO_2$ characteristics down to the cruising altitudes. Therefore, we would like to keep the wind vectors at 250 hPa, the pressure surface corresponding to the typical cruising altitude. We have added the following sentence to mention the contribution by Chandra et al. (2017):

"The high $CH_4$ values in the Asian summer monsoon anticyclone, its formation mechanism, and outflow from the anticyclone were recently discussed by Chandra et al. (2017)." (P11 L5)

*Figure 3 and 4. The histogram panel looks too messy. The author can consider 10o latitudinal band instead of 5o for plotting histogram.*

We thank the referee for the suggestion. However, we would like to keep the histogram as is for the following reasons. As discussed in the text, the shape of the histograms to some degree reflects the nature of the $CO_2$ variations, like the spreading of the histograms found over boreal Eurasia in the summer and those around Japan in the spring. We agree that the histograms are bit "messy" due in some cases to limited number of measurement flights (i.e. sampling bias). However, one of the objectives of this manuscript is to disclose the full extent, graphically at least, of the currently available dataset. To this end, we think it is important to display as much as possible the density of the data points in space and time (and data at which locations could be sampling biased). In the future, we will make measurement data available in a different format for data users, but we think it is good that some parts of the available data are also visible in the current manuscript.

---

## Editor Decision (ED1)

Dear Taku Umezawa,

Thank-you for the work that you and your co-authors have undertaken to address the reviewers comments on your manuscript. Overall I think you have answered the concerns sufficiently. In reading the manuscript again, I would like to recommend the following technical corrections. The page and line numbers refer to the version of the manuscript which shows the tracked changes.

P1, line 20: You appear to have misunderstood the correction requested by reviewer 1 and have deleted 'the' earlier in the sentence than was required. The sentence should read '… observed in the vertical profiles of $CO_2$ over the Pacific Rim of continental East Asia.'

P1, line 23: delete 'the' in two places so that the sentence reads '…over the northwestern Pacific comes from continental East Asian emissions in spring, …'

P1, line 25: add 'a' before 'distinct imprint'

P4, line 7: delete 'the' before 'North America'

P7 line 21 onwards. Reviewer 1 has suggested the location of airports near cities as one explanation for the difference between the Asian profiles in summer compared to those from North America, and your additional paragraph discussing this is useful. I wonder whether it would also be appropriate to mention here that coastal/inland differences may also be important. I think most of the North American sites are inland, and are consistent with the features you see at DEL which is also inland. The other Asian sites are coastal, and may be influenced by sea-breezes in summer which could also reduce the signal from biospheric uptake.

Regards,

Rachel Law, rachel.law@csiro.au

---

## Author Response (AR2)

Author's response by Taku Umezawa

Dear Editor Dr. Law,

Thank you for your suggestions. Our responses are decribed below.

*Thank-you for the work that you and your co-authors have undertaken to address the reviewers comments on your manuscript. Overall I think you have answered the concerns sufficiently. In reading the manuscript again, I would like to recommend the following technical corrections. The page and line numbers refer to the version of the manuscript which shows the tracked changes.*
*P1, line 20: You appear to have misunderstood the correction requested by reviewer 1 and have deleted 'the' earlier in the sentence than was required. The sentence should read '... observed in the vertical profiles of CO2 over the Pacific Rim of continental East Asia.'*
*P1, line 23: delete 'the' in two places so that the sentence reads '...over the northwestern Pacific comes from continental East Asian emissions in spring, ...'*
*P1, line 25: add 'a' before 'distinct imprint' P4, line 7: delete 'the' before 'North America'*
Thank you for your thorough corrections. We have made corrections for all the above places according to the suggestions.

*P7 line 21 onwards. Reviewer 1 has suggested the location of airports near cities as one explanation for t he difference between the Asian profiles in summer compared to those from North America, and your additional paragraph discussing this is useful. I wonder whether it would also be appropriate to mention here that coastal/inland differences may also be important. I think most of the North American sites are inland, and are consistent with the features you see at DEL which is also inland. The other Asian sites are coastal, and may be influenced by sea-breezes in summer which could also reduce the signal from biospheric uptake.*
We thank the editor for the important suggestion about the coastal/inland differences. All of our measurement sites, except DEL, are indeed in coastal areas and under

considerable influence of maritime air with a limited imprint of terrestrial uptake. We have added the following sentence (underlined) in the paragraph to concisely adopt your suggestion. More detailed discussion, including influence of sea breeze, needs further analysis of data for individual sites, given that even the Asian coastal sites presented here have variety of meteorological conditions.

[revised manuscript text omitted]